# CXCR4: From Signaling to Clinical Applications in Neuroendocrine Neoplasms

**DOI:** 10.3390/cancers16101799

**Published:** 2024-05-08

**Authors:** David Sanchis-Pascual, María Isabel Del Olmo-García, Stefan Prado-Wohlwend, Carlos Zac-Romero, Ángel Segura Huerta, Javier Hernández-Gil, Luis Martí-Bonmatí, Juan Francisco Merino-Torres

**Affiliations:** 1Endocrinology and Nutrition Department, University and Politecnic Hospital La Fe (Valencia), 46026 Valencia, Spain; delolmo_margar@gva.es (M.I.D.O.-G.); merino_jfr@gva.es (J.F.M.-T.); 2Joint Research Unit on Endocrinology, Nutrition and Clinical Dietetics, Health Research Institute La Fe, 46026 Valencia, Spain; 3Nuclear Medicine Department, University and Politecnic Hospital La Fe (Valencia), 46026 Valencia, Spain; prado_ste@gva.es; 4Patholoy Department, University and Politecnic Hospital La Fe (Valencia), 46026 Valencia, Spain; zac_carrom@gva.es; 5Medical Oncology Department, University and Politecnic Hospital La Fe (Valencia), 46026 Valencia, Spain; segura_ang@gva.es; 6Instituto de Tecnología Química, Universitat Politècnica de València, Consejo Superior de Investigaciones Científicas, 46022 Valencia, Spain; jahergi@upvnet.upv.es; 7Medical Imaging Department, Biomedical Imaging Research Group, Health Research Institute, University and Politecnic Hospital La Fe, 46026 Valencia, Spain; luis.marti@uv.es; 8Department of Medicine, University of Valencia, 46010 Valencia, Spain

**Keywords:** CXCR4, CXCL12, cancer, neuroendocrine neoplasm, radiotracer

## Abstract

**Simple Summary:**

Neuroendocrine neoplasms are a heterogeneous group of malignant tumors that originate from the diffuse endocrine system. They generally have a slow course and somatostatin receptor-targeted based management is the first line of treatment. However, high-grade tumors and neuroendocrine carcinomas have a poor prognosis and somatostatin receptor-targeted therapy is not effective. The membrane receptor CXCR4 has been studied in several neoplasms and it is known to be overexpressed in aggressive tumors and associated with a worse prognosis. However, there is a lack of evidence of its use in neuroendocrine neoplasms. For that reason, this review describes the significance of CXCR4 and its possible clinical applications in the diagnostic and therapeutic management of neuroendocrine neoplasms.

**Abstract:**

There are several well-described molecular mechanisms that influence cell growth and are related to the development of cancer. Chemokines constitute a fundamental element that is not only involved in local growth but also affects angiogenesis, tumor spread, and metastatic disease. Among them, the C-X-C motif chemokine ligand 12 (CXCL12) and its specific receptor the chemokine C-X-C motif receptor 4 (CXCR4) have been widely studied. The overexpression in cell membranes of CXCR4 has been shown to be associated with the development of different kinds of histological malignancies, such as adenocarcinomas, epidermoid carcinomas, mesenchymal tumors, or neuroendocrine neoplasms (NENs). The molecular synapsis between CXCL12 and CXCR4 leads to the interaction of G proteins and the activation of different intracellular signaling pathways in both gastroenteropancreatic (GEP) and bronchopulmonary (BP) NENs, conferring greater capacity for locoregional aggressiveness, the epithelial–mesenchymal transition (EMT), and the appearance of metastases. Therefore, it has been hypothesized as to how to design tools that target this receptor. The aim of this review is to focus on current knowledge of the relationship between CXCR4 and NENs, with a special emphasis on diagnostic and therapeutic molecular targets.

## 1. Introduction

### 1.1. Role of CXCL12 and CXCR4

Chemokines are a group of small molecules (~8–10 k Da) that belong to the cytokine family together with angiogenic factors, growth factors, or interferons and are secreted not only by neoplastic cells but also by macrophages, lymphocytes, or dendritic cells. Their main function is to stimulate chemotaxis of immune system cells as part of the inflammatory response through the interaction with fibroblasts or endothelial cells, while in neoplastic status they induce angiogenesis and sustain cell growth [1]. This action is exerted by binding the N-terminal domain, which is rich in the amino acid cysteine, to its specific receptor [2]. Depending on the distribution of this amino acid, four subtypes of cytokines are identified: CXC, CX3C, CC, and C [3].

Among the 50 types of chemokines known today, the chemokine CXCL12, which is also recognized as stromal cell-derived factor-1 (SDF-1) [4], has some characteristics that make it different from the rest of its family. Firstly, it is the only cytokine whose mRNA can be subjected to a process known as differential splicing, which is why up to six variants of this molecule have been recognized in humans (α to ϕ) [5,6]. Secondly, it is a chemokine with an almost exclusive affinity for a single receptor, with nothing to do with the promiscuity of the rest of the cytokines [7]. Until a few years ago, CXCR4 was recognized as the only natural receptor for CXCL12, although it has been discovered that it can also mediate its action through interaction with the atypical chemokine receptor type 3 (ACKR3), previously known as chemokine C-X-C motif receptor 7 (CXCR7) [8,9]. 

CXCL12 is probably the most important cytokine that binds to CXCR4 but this receptor does not only bind to this type of molecule. Different ligands for CXCR4 have been recognized, most notably macrophage inhibitory factor (MIF) [10] and ubiquitin [11,12].

### 1.2. Structure and Signaling Pathway of the CXCL12-CXCR4-ACKR3 Axis

CXCR4 is a molecular structure that has also presented different names throughout history. It was initially called leukocyte-derived seven-transmembrane receptor (LESTR) when it was isolated from a human blood monocyte cDNA library [13]. It has also been known as cluster of differentiation 184 (CD184) or fusin. The latter name refers to the ability of the human immunodeficiency virus 1 (HIV-1) to infect human cells by the process of fusion following the binding of its glycoprotein 120 (gp120) [14]. Although its natural ligand is the chemokine CXCL12 (as mentioned before), there is greater evidence that it has a wider spectrum of interactions with other molecules. In fact, it also recognizes ligands as small proteins like ubiquitin or the macrophage migration inhibiting factor (MIF) [15,16]. This receptor belongs to the family of G protein-coupled receptors (GPCRs), which are characterized by the presence of seven membrane-spanning α-helical segments separated by alternating intracellular and extracellular loop regions [17]. The intracytoplasmic domain of the receptor remains in contact with a heterotrimeric G protein that is composed of a G_α_, G_β_, and G_γ_ subunits and, when the interaction between CXCL12 and CXCR4 occurs, the exchange of guanosine diphosphate (GDP) for triphosphate (GTP) leads to a complex process in which a GTP-bound G_α_ monomer and a G_βγ_ dimer are released [18].

The G_α_ subunit produces an inhibition of the adenylate cyclase leading to an increase in the intracellular calcium (Figure 1) mediated by the decrease in the concentration of adenosine 3′,5′-cyclic monophosphate (cAMP). It also interacts directly with the Src family of tyrosine kinases and then activates the signaling pathway of MEK1/2-Erk1/2 [19]. The G_βγ_ subunit activates phosphatidyl-inositol-3-OH kinase (PI3K) and consequently generates an increase in phosphatidylinositol triphosphate (PIP3), while the interaction with phospholipase C generates diacylglycerol (DAG) and inositol-(1,4,5)-triphosphate (IP3). IP3 increases intracellular calcium deposition after outflow from the endoplasmic reticulum (ER), while DAG interacts with protein kinase C and mitogen-activated protein kinase (MAPK) [20].

When CXCL12 binds ACKR3/CXCR7, a different signaling pathway is developed because of the biochemical difference between classical and atypical chemokine receptors, which basically boils down to the fact that atypical cytokine receptors (ACKRs) lack G proteins and its effects are calcium-independent [21]. The signal pathway through β-arrestin proteins becomes the main way the ACKR3 activation leads to its tumorigenic properties. β-arrestins increase the MEK/ERK axis and the protein kinase B (also known as Akt) activity [22]. The binding of CXCR4 to its agonist ligand results in phosphorylation and internalization of the receptor [23,24]. However, once inside the cell, it can be recycled and transported back to the plasma membrane or it can be degraded in the cell lysosomes [25]. The first scenario occurs in a PKC-mediated phenomenon [26], whereas the second case takes place after interaction with E3 ubiquitin ligase [27].

## 2. CXCR4 and Cancer

Firstly, the involvement of CXCR4 as a co-receptor in HIV infection overshadowed its potential as a tumorigenesis-related agent and it was not until 1999 when Burger et al. noticed that this protein favored migration of B cells in chronic lymphocytic leukemia. Since then, the link between CXCR4 and tumoral disease has been reviewed and, for instance, the implication of CXCR4 in more than 23 cancers is well known [28,29,30]. Considering that CXCR4 functions involve the promotion of cell growth, proinflammatory cell recruitment, angiogenesis, and cell migration, it is not surprising that the pathological activation of this receptor favors the development of tumoral disease. To be more accurate, the hyperactivation of the CXCL12/CXCR4/AKR3 axis is associated with increased tumor size, lower degree of cell differentiation, higher probability of recurrence, worse response to chemotherapy, and decreased overall survival [31,32]. The role it plays in cell growth and its different effects on stromal tissue have placed this receptor in the spotlight of the scientific community. CXCR4 has been studied in practically all the different types of cancer because its expression is independently associated with decreased survival [33]. In fact, it is being investigated as to whether it could be a pan-cancer marker of the microenvironment status [34].

The presence of metastases drastically worsens cancer prognosis and CXCR4 is closely related to this phenomenon in various solid tumors. It is hypothesized that the upward adjustment of the CXCR4/CXCL12 axis occurs in organs on which metastases frequently settle such as the liver, lung, brain, or bone [5,35,36] and this fact can be ratified if it is taken into account that the blockade of this axis leads to metastatic dissemination in animal studies [37,38]. Regarding the possible underlying mechanisms, the influence on the epithelial–mesenchymal transition (EMT) is postulated. This is a process characterized by the disarticulation of tight junctions and loss of apicobasal polarity [39] that facilitates distant dissemination and invasion of different organs by the acquisition of a mesenchymal phenotype. This process involves interleukin 11 [40], the NF-kB receptor [41], and CXCR4 [42,43].

Lastly, CXCR4 is closely related not only to solid tumors but also to the hematopoietic system [44]. Such is the case that the CXCR4/CXCL12-knockout mice exhibit specific characteristics which consist of heart malformations, abnormal cerebellar development, and absence of myelopoiesis and B lymphopoiesis [45,46,47]. This phenomenon can be explained if we take into consideration that CXCL12 is one of the most relevant cytokines involved in the chemotactic response of hematopoietic stem cells (HSCs) [48]. Having this ligand-receptor axis intact results necessary not only for the homing of HSCs through the bone marrow but also in retaining them in the hematopoietic microenvironment [49,50]. This knowledge has led to the development of strategies that target this level, such as the CXCR4 antagonist plerixafor, which is used in bone marrow transplant in patients with multiple myeloma or non-Hodgkin lymphoma due to its ability to mobilize HSCs from the bone stroma to the peripheral blood [51].

## 3. CXCR4 and NENs

### 3.1. Introduction to Neuroendocrine Neoplasms

NENs are a heterogeneous group of malignant tumors whose origin relies in the cells of the diffuse endocrine system, which are scattered throughout the human body, although the most frequent locations are in the gastrointestinal (GI) tract or in the lung. The incidence of NENs varies substantially according to the location of the primary tumor, being approximately 3.56 new cases per 100,000 in gastroenteropancreatic NENs (GEP-NENs), 1.49/100,000 in bronchopulmonary NENs (BP-NENs), and 0.84/100,000 in unknown primary NENs [52]. It is important to highlight the association of NENs with genetic syndromes such as multiple endocrine neoplasia syndrome type 1 [53]. NENs can be classified depending on whether they produce biologically active substances or not. Currently, it is considered that about 60% of NENs are non-functioning [54]. Carcinoid syndrome is the most common of the many syndromes that can develop due to hormone production [55] (such as insulinoma, glucagonoma, and gastrinoma).

The expression of somatostatin receptors (SSTR) on the cell membrane is a typical feature of NENs and it has diagnostic as well as therapeutic approaches [56]. In fact, the ability to diagnose NENs has improved substantially thanks to the incorporation of gallium-68(68Ga)-labeled DOTA tracers, such as DOTA-TOC, DOTA-TATE, and DOTA-NOC because of their sensitivity and specificity that reach 97% and 92%, respectively [57], compared to Indium-111 scintigraphy (sensitivity 72% and specificity 92%). 

Somatostatin analogs (SSA) constitute the first line treatment in NENs due to an antisecretory as well as an antiproliferative effect. In fact, administration of both octreotide [58] or lanreotide [59] has demonstrated increases in progression-free survival (PFS) versus placebo (14.3 vs. 6.0 months, HR 0.34, and >27 vs. 18 months, HR 0.47, respectively) in GEP-NENs. Regarding BP-NENs, only lanreotide has shown benefits in PFS [60] (16.6 vs. 13.6 months, HR 0.90). There are five types of SSTRs, although the drugs currently available focus on SSTR2A and SSTR5 agonism [61]. Lanreotide and octreotide mainly stimulate SSTR2 while pasireotide exerts its action after binding to SSTR2 and SSTR5. In general, NENs are indolent and slow-growing tumors. The main prognostic factor for GEP-NENs is the tumor grade according to the latest WHO classification, which takes into account cytologic features, the number of mitoses per field, and the Ki-67 proliferation index [62]. BP-NENs are governed by a similar classification but this does not take into account the proliferation index but the presence of necrosis on histology [63]. Four variants can be recognized: typical carcinoid (TC), atypical carcinoid (AC), large-cell neuroendocrine carcinoma (LCNC), and small-cell neuroendocrine carcinoma (SCNC).

### 3.2. Implications of CXCR4 Expression in NENs

The molecular study of NENs involves the detection and evaluation of multiple membrane targets, among which SSTRs are the most important. The more the SSTR is expressed (especially SSTR2A), the lower the grade and therefore the better the prognosis of NENs [64,65,66,67,68,69]. As with other types of neoplasms, the chemokine receptor CXCR4 is becoming increasingly relevant to researchers in the field of NENs. Circelli et al. demonstrated that the PI3K/Akt/mTOR pathway is enhanced both in GEP-NENs and in BP-NENs cell lines throughout an upregulation of the CXCR4-CXCL12 axis [70]. Indeed, the hyperactivation of this intracellular pathway has led to the development of mTOR inhibitors for the treatment of pancreatic NENs [71,72].

Among the multiple factors that determine the functioning of the CXCL12-CXCR4 axis, the hypoxia phenomenon plays a fundamental role in carcinogenesis [73,74,75] and in the homeostasis of these molecules [76,77]. The hypoxia-inducible factors 1α and 2α (HIF1α and HIF2α) increase the expression of CXCR4, confer greater aggressiveness, and result in lower survival in patients with ileal NENs [78,79]. Kaemmerer et al. demonstrated an inverse relationship between CXCR4 expression and overall survival (OS) in GEP-NENs, since those patients with a marked expression of this receptor had a lower, although not significant, OS compared to those with a weak expression (34.0 vs. 50.0 months, *p* = 0.068). This expression was higher in high-grade tumors (differentiating also between grades 3a and 3b) versus low-grade tumors. Furthermore, a positive correlation was identified between CXCR4 and Ki-67 index (r 0.39; *p* < 0.001) and with SSTR5 expression (r 0.27; *p* = 0.003), while SSTR2A expression showed a robust inverse correlation (r −0.50; *p* < 0.001) [80]. 

No clear agreement exists about the relationship between CXCR4 expression and GEP-NEN location. While Mai et al. showed higher expression in those whose primary tumor is located in the appendix or colon (*p* = 0.024) [81], Popa et al. showed no statistical differences among them, but greater expression in colonic primary tumors but less immunoreactive intensity in appendix ones [82]. Interestingly, no statistical differences were found in both studies between primary tumors and metastases in the intensity of expression. In relation to hormone production, it appears that expression is higher in those non-functioning neoplasms (*p* = 0.019) [81]. Regarding BP-NENs, an inverse correlation with OS has also been shown. TC and AT tend to show lower expression of CXCR4 but are high and intense in SCLC [83]. The role that CXCR4 plays in the EMT is crucial in the pathogenesis of metastatic disease in both GEP and BP-NENs [84,85,86,87]. It seems that among multiple target organs, bone involvement is intimately influenced by the overexpression of this receptor, in both in vitro [88] and in vivo [89] studies. 

### 3.3. CXCR4 as a Target for Imaging Diagnosis on NENs

The use of computed tomography (CT) and magnetic resonance imaging (MRI) scans is essential in the diagnosis and staging of tumoral disease. However, the use of functional imaging techniques through the administration of radiotracers has become a cornerstone in the management of patients with NENs. The main molecular targets in the study of NENs are somatostatin receptors, especially SSTR2 and SSTR5. The first imaging techniques that emerged were 111In-DTPA-Pentetreotide (Octreoscan^®^) and 99mTC-EDDA-HYNIC-Thr3-octreotide (Tektrotyd^®^) scintigraphy with improved spatial resolution using single photon emission tomography (SPECT/CT). Diagnostic performance was subsequently increased with the introduction of radiopharmaceuticals suitable for positron emission tomography (PET/CT) such as 68Ga-labeled tracers. However, the expression of SSTRs decreases with the increasing tumor grade, which influences the 68Ga-PET/CT sensitivity (72.2% in G1 vs. 40.8% in G3 NENs) and maximum standardized uptake value (SUVmax) (29.2 in G1 vs. 12.8 in G3 NENs) [90], so SSTR targeting may be less useful in the diagnosis and follow-up of dedifferentiated NENs. In this scenario, ^18^F-fluorodeoxyglucose (18F-FDG) PET/CT provides additional information and insight into the metabolic state of the neoplastic lesions [91,92]. However, this technique is not free of interferences that may hinder its correct interpretation [93] and it is therefore necessary to investigate alternative molecular targets for lesions with a lower degree of differentiation.

In 2008, Uy et al. developed the drug plerixafor (AMD3100), a CXCR4 antagonist that prevents binding of its ligand CXCL12/SDF-1 and is used for stem cell mobilization from bone marrow prior to hematopoietic progenitor transplantation [94]. In 2014, Aghanejad et al. developed a 68Gallium-plerixafor radiotracer that demonstrated its potential utility in the field of Oncology by detecting breast cancer cells in an in vivo mouse model [95]. However, previously, Gourni et al. designed a molecule composed of a cyclic peptide CPCR4-2 labeled with 68Ga (cyclo(D-Tyr1-[NMe]-D-Orn2-[4-(aminomethyl) benzoic acid,68Ga-DOTA]-Arg3-2-Nal4-Gly5, also known as pentixafor), which has revealed higher specificity for CXR4 and greater in vivo stability for the study of malignant neoplasms in humans [96,97,98]. 68Ga-Pentixafor seems to be an interesting future tool and therefore studies are being carried out to show the usefulness of this in various types of neoplasms [99].

Regarding NENs, Werner et al. were the first to noninvasively evaluate CXCR4 expression by 68Ga-Pentixafor PET/CT compared with 68Ga-DOTA-TOC and 18F-FDG PET/CT in 12 patients with GEP-NENs. 68Ga-Pentixafor was negative in all G1-NENs while 68Ga-DOTA-TOC and 18F-FDG PET/CT identified lesions in 12/12 and 11/12 patients, respectively. In G2-NENs, 68Ga-Pentixafor was positive in half of the cases (2/4) whereas both 68Ga-DOTA-TOC and 18F-FDG were positive in all of them and in G3-NENs both 68Ga-Pentixafor and 68Ga-DOTA-TOC confirmed positivity in four out of five patients when 18F-FDG was positive in five out of five of the cases. These data agree with what has been published to date on the lower expression of SSTR and the increase in CXCR4 at higher tumor grade. However, the results further support the use of 18F-FDG against direct targeting of CXCR4 with pentixafor. In addition, the number of lesions identified was markedly lower compared to the other radiotracers, both overall (69 lesions for 68Ga-Pentixafor, 127 for 18F-FDG, and 245 for 68Ga-DOTA-TOC) and stratified by WHO tumor grading [100]. Interestingly, not only are there differences in the ability to detect lesions in different patients but intraindividual variability has also been shown in which some G3-NENs lesions may be positive for 18F-FDG and negative for 68Ga-Pentixafor or vice versa. That makes the management of patients with NENs more complex because of the existence of multiple lesions with different molecular behaviors [101]. 

To evaluate the diagnostic potential of CXCR4 labeling in dedifferentiated tumors, Weich et al. confronted 18F-FDG and 68Ga-Pentixafor PET/CT in 11 patients newly diagnosed from GEP-NEC and studied IHC expression of CXCR4. In a per-patient analysis, 18F-FDG-avid lesions were detected in all patients while 68Ga-Pentixafor was positive in 10/11 patients. In a per lesion analysis, the ability of 18F-FDG to reveal more lesions in comparison with 68Ga-Pentixafor was shown (102 vs. 42 lesions, *p* < 0.001). In relation to radiotracer uptake intensity, 18F-FDG showed a higher SUVmax in contrast to 68Ga-Pentixafor (12.8 ± 9.8 vs. 5.2 ± 3.7, *p* < 0.001) and greater tumor-to-background ratios (TBR) (7.2 ± 7.9 vs. 3.4 ± 3.0, *p* < 0.001). With respect to IHC, the overall CXCR4 expression was cataloged as low in 7/11 patients and there was no correlation between the intensity of CXCR4 expression and the 68Ga-Pentixafor uptake in biopsies [102].

With regard to BP-NENs, the correlation between the 68Ga-Pentixafor uptake in PET/CT images and the CXCR4 expression by mean fluorescence index and IHC was studied. Although there was an increased uptake in all patients, no correlation was found to both cytologic features [103]. However, a study comparing the usefulness of 68Ga-Pentixafor versus other types of radiotracers in BP-NENs has not yet been developed. In addition, 68Ga-Pentixafor has not shown an association with clinical parameters such as OS or PFS both in GP-NENs and SCLC, although it does appear to be related to leukocyte and platelet counts [104].

Among the multiple interrelated processes in CXCR4 homeostasis, the Wnt/β-catenin molecular pathway is fundamental in the correct functioning of the CXCL12-CXCR4 axis [105,106] and is also deregulated in about 25% of patients with GEP, lung, or thymus NENs [107]. For this reason, the possibility of modulating the expression density of CXCR4 and its 68Ga-Pentixafor binding capacity in NEN cell lines has been studied, achieving promising results that open the door to future studies with Wnt inhibitors or activators [108].

### 3.4. CXCR4 Targeting as Treatment of NENs

Precision medicine consists of individualizing treatment according to the specific characteristics of each patient and neoplasm. In the case of NENs, a good example is treatment with SSA, which performs its action specifically against neoplastic cells that express SSTR in their plasma membrane. However, sometimes, this treatment is not sufficient and it is necessary to identify new therapeutic targets. CXCR4 emerges as a possible target and the selective approach against it can be carried out using different therapeutic strategies [109] (Figure 2).

#### 3.4.1. Synthetic Peptides

Administration of TF14016, a direct CXCR4 inhibitor, has been shown in animal models to decrease the number and size of pulmonary metastases in SCLC. In addition, a lower expression of vascular endothelial cell growth factor was recorded [110]. A cyclic peptide antagonist called LY2510924 was studied in a phase II trial in patients with SCLC added to carboplatin/etoposide but did not show efficacy (PFS 5.88 vs. 5.85, *p* = 0.9806) although its toxicity profile was acceptable [111]. Although there is limited information about the treatment of NENs with this type of molecules, there are peptides such as balixafortide, motixafortide, and mavorixafor that have been studied in the treatment of solid neoplasms [112,113,114], HSC mobilization prior to bone marrow transplantation [115,116], and even in rare warts, hypogammaglobulinemia, immunodeficiency, and myelokathexis (WHIM) syndrome [117] but not yet neither in GEP nor BP-NENs.

#### 3.4.2. Monoclonal Antibodies

Several antibodies against CXCR4 have been studied, although most trials are in the early stages and evidence in NENs is limited [118]. For the time being, no in vivo studies have been developed and the information available comes from in vitro studies. The effect of ulocuplumab (a monoclonal antibody that prevents CXCL12 binding) has been studied in pancreatic NENs [119]. Although it has not been shown to exert a cytolytic effect on tumor cells, a reduced migration toward the liver and bone by inhibiting EMT has been observed. Intriguingly, Yingnan Si et al. developed dual SSTR2/CXCR4 targeted extracellular vesicles-delivered combined therapy through monoclonal antibodies against pancreatic, thyroid, and lung NENs [120]. This experimental treatment showed an anticancer efficacy both in vitro and in vivo models and no systemic toxicity was reported. 

#### 3.4.3. Peptide Receptor Radionuclide Therapy

Theragnosis is a medical approach combining diagnosis and therapy to tailor treatment strategies for individual patients, primarily used in cancer care to identify specific receptors and then target them with precise radiotracer. 

As mentioned above, plerixafor is a CXCR4 antagonist mainly used in hematopoietic stem cell transplants. However, it has also been studied for stem cell collection in patients with a NEN massive bone marrow infiltration, prior to the administration of 177Lu-DOTATATE, and initiated after the failure of a granulocyte-colony stimulating factor [121]. NENs are a type of tumor in which peptide receptor radionuclide therapy (PRRT) has been implemented since the publication of the trial NETTER-1 in 2017 [122]. This trial showed the superiority of 177Lu-DOTATATE versus SSA high-dose monotherapy in terms of PFS. Although a non-significant improvement in OS was subsequently identified (36.3 months vs. 40 months in the PRRT-Lu arm, *p* = 0.30), this effect was attributed to the high rate (36%) of cross-over of patients in the control arm to PRRT after progression [123].

The role of 68Ga-Pentixafor in the diagnosis of high-grade NENs and dedifferentiated NECs has been investigated. However, due to its altered affinity for CXCR4 when interacting with metal-chelate conjugates and its relatively fast clearance [124], 68Ga-Pentixafor does not appear to be a valid tool for the therapeutic management of malignancies. Thus, Schottelius et al. designed a novel molecule with improved pharmacokinetics called pentixather, which was labeled with 177Lu [125]. Most of the available evidence for this novel radiopharmaceutical comes from its endoradiotherapeutic use in hematologic malignancies. It has been shown to elicit high responses and decrease 18F-FDG uptake in multiple myeloma lesions both bound to 177Lu and 90Y [126,127]. It has also demonstrated utility in refractory acute leukemia and diffuse large-cell lymphoma [128,129] and may be useful in glioblastoma cells [130]. The available evidence for pentixather in the treatment of NENs comes only from BP-NENs in animal studies. On the one hand, 177Lu-Pentixather has been shown to decrease tumor growth and increase OS in mice with SCLC [131]. On the other hand, the administration of 212Pb-Pentixather associated with a thioredoxin reductase inhibitor caused a delay in tumor growth in mice with SCLC xenograft [132].

## 4. Conclusions

CXCR4 and its ligand CXCL12 are essential in the tumorigenesis and development of NENs. It appears that SSTRs and CXCR4 maintain an antagonistic relationship that favors the latter in high-grade, dedifferentiated, and metastatic tumors. Consequently, current research is focusing on selectively targeting this membrane receptor. At the moment, it seems that targeted diagnosis using ^68^Ga-Pentixafor does not provide more information than ^18^F-FDG although there are mechanisms that may influence its uptake and be relevant in the future. The treatment of NENs with molecules specifically directed against CXCR4 is in the preclinical phase, although the data on radiopharmaceuticals such as ^177^Lu-Pentixather or ^212^Pb-Pentixather in the theragnosis treatment of BP-NENs are encouraging. 

## Figures and Tables

**Figure 1 cancers-16-01799-f001:**
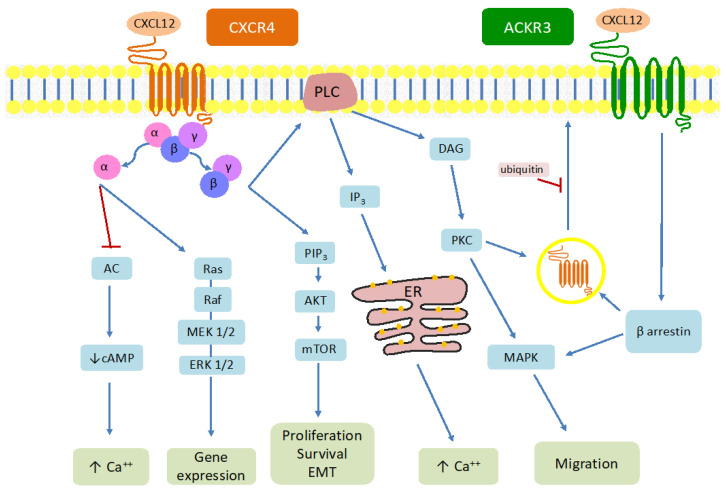
Representation of the signaling pathway in the activation of CXCR4 (left) and ACKR3 (right). Blue arrows mean activation while red arrows represent metabolic pathway inhibition. Note that G proteins and calcium do not participate as second messengers after binding CXCL12 to ACKR3. Acronyms: C-X-C motif chemokine ligand 12 (CXCL12), chemokine C-X-C motif receptor 4 (CXCR4), atypical cytokine receptor type 3 (ACKR3), protein kinase C (PKC), adenylate cyclase (AC), adenosine 3′,5′-cyclic monophosphate (cAMP), extracellular signal-regulated kinases (ERK), mitogen-activated protein kinase (MAPK), diacylglycerol (DAG), inositol-(1,4,5)-triphosphate (IP3), phosphatidylinositol triphosphate (PIP3), phospholipase C (PLC), mammalian target of rapamycin (mTOR), endoplasmic reticulum (ER).

**Figure 2 cancers-16-01799-f002:**
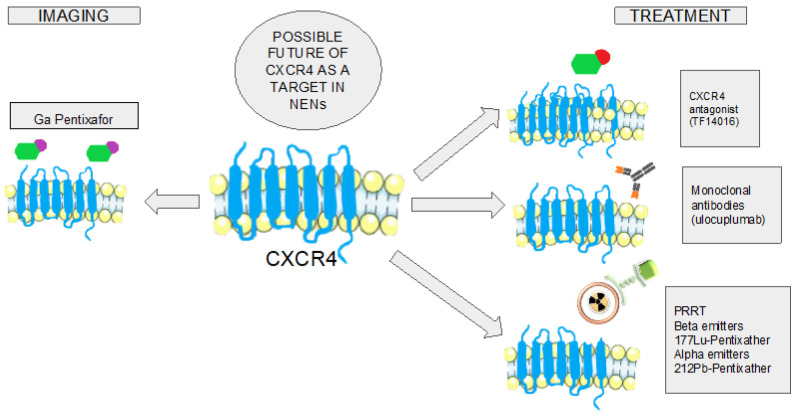
Schematic representation that summarizes the future of CXCR4 as a target in NENs, in both diagnostic and therapeutic approaches. Acronyms: peptide receptor radionuclide therapy (PRRT), Gallium (Ga), Lead (Pb), and Lutetium (Lu).

## Data Availability

Data are contained within the article.

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
