# Peer review of "CXCR4: From Signaling to Clinical Applications in Neuroendocrine Neoplasms"

_cancers, 2024, doi:10.3390/cancers16101799_

Round 1

Reviewer 1 Report

Comments and Suggestions for Authors

The aim of the work is to take stock of neuroendocrine oncological pathologies. Extremely learned paper in terms of oncological biochemistry for the detailed descriptions of the interactions between CXCL12 and CXCR4 with the recent history and the other receptors capable of binding to CXCR4. The biochemistry of this interaction helps to explain the chemotaxis of the immune system, the inflammatory response and angiogenesis in neoplasia. The description continues with the functions of CXCR4 involving the promotion of cell growth, recruitment of proinflammatory cells, angiogenesis and cell migration, it is not surprising that pathological activation of this receptor promotes the development of tumor disease. The same goes for metastases. As with other types of malignancies, the chemokine receptor CXCR4 is becoming increasingly relevant to researchers in the field of NENs. Neoplasms of the neuroendocrine system ubiquitous in all tissues and frequent in the gastrointestinal system, lung, pancreas. Capable of secreting hormones which often reveal their presence but which can also produce imperfect hormones and therefore incapable of having effects on the relevant receptors. On the surface of the cells there are somatostatin receptors which, among other things, has been and is used for therapeutic control. For this purpose we point out that somatostatin and analogues can have consequences by acting on the motility of the gallbladder (PMID: 38051513 to be cited for completeness in the bibliography).

We absolutely agree that the main diagnostic aids are the CT scan for the characteristic appearance of the lesions and the gallium PET scan. The article concludes with the possibility of preparing a personalized treatment by modulating the expression density of CXCR4 and its binding capacity with 68Ga-Pentixafor in NEN cell lines. Written in excellent English with a substantial bibliography that supports the initial endpoint with excellent iconography

Author Response

Dear reviewer,

Thank you very much for taking the time to review this manuscript.

Please find the article you mentioned is quoted on line 183 and you will find it underlined in yellow color in the re-submitted file.

Best regards,

David Sanchis.

Reviewer 2 Report

Comments and Suggestions for Authors

This review manuscript covers the CXCR4 as a potential new target for neuroendocrine neoplasms (NENs) for both diagnostic and therapeutic purposes. It is well addressed of the natural ligands CXCL12 and its signal-axis with CXCR4/ACKR3 which involves in multiple facets from the angiogenesis to tumor metastases of multiple solid tumors and hematological malignancies. In the later part of the review, it describes about the updates of the synthetic CXCR4 ligand analogues in preclinical and clinical applications. But as the author concluded, 68Ga-Pentixafor, as a PET/CT imaging radiotracer didn't show more favorable result than 18F-FDG, 68Ga-DOTATOC, or IHC regarding the diagnosis of GEP-NEC, or BP-NEN. And the treatment using 177Lu-Pentixather or 212Pb-Pentixather has encouraging results but has not been widely applied in NENs.

Minor revises or more supporting needed:

1. In the legend of Figure 2, some of the acronyms were redundant, like CXCR4, and NENs. Also, Pb should be lead, and not Plumber.

2. Please address the significant benefit of CXCR4 as the theranostic target in GEP-NETs and BP-NENs as compared to SSTR2 target, especially from the aspects of the available analogues, preclinical and clinical study outcome.

Author Response

Dear Reviewer,

Thank you very much for taking the time to review this manuscript. Please find the detailed response and the corresponding correction highlighted in the re-submitted file.

Reviewer 3 Report

Comments and Suggestions for Authors

In my opinion this is a nice review and I actually have no comments.

Comments on the Quality of English Language

Minor editing of English language necessary. Please check for spelling errors.

Author Response

Dear reviewer,

Thank you very much for taking the time to review this manuscript.

Please find the corrections I have made are underlined in yellow in the re-submitted file.

Best regards,

David Sanchis.
